# Individualized Immunological Data for Precise Classification of OCD Patients

**DOI:** 10.3390/brainsci8080149

**Published:** 2018-08-09

**Authors:** Hugues Lamothe, Jean-Marc Baleyte, Pauline Smith, Antoine Pelissolo, Luc Mallet

**Affiliations:** 1Centre Hospitalier Intercommunal de Créteil, 94000 Créteil, France; lamothehugues@gmail.com (H.L.); jean-marc.baleyte@chicreteil.fr (J.-M.B.); 2Institut du Cerveau et de la Moelle Epinière, Sorbonne Universités, UPMC Univ Paris 06, CNRS, INSERM, 75013 Paris, France; pauline.hh.smith@gmail.com; 3Fondation FondaMental, 94000 Créteil, France; a.pelissolo@gmail.com; 4Assistance Publique-Hôpitaux de Paris, Pôle de Psychiatrie, Hôpitaux Universitaires Henri Mondor—Albert Chenevier, Université Paris-Est Créteil, 94000 Créteil, France; 5INSERM, U955, Team 15, 94000 Créteil, France; 6Department of Mental Health and Psychiatry, Global Health Institute, University of Geneva, 1202 Geneva, Switzerland

**Keywords:** psychiatry, OCD, obsessive–compulsive disorder, Tourette syndrome, immunology, cytokines, pediatric autoimmune neuropsychological disorders associated with streptococcal infection (PANDAS), pediatric acute-onset neuropsychiatric syndrome (PANS), *Toxoplasma gondii*, *Streptococcus pyogenes*

## Abstract

Obsessive–compulsive disorder (OCD) affects about 2% of the general population, for which several etiological factors were identified. Important among these is immunological dysfunction. This review aims to show how immunology can inform specific etiological factors, and how distinguishing between these etiologies is important from a personalized treatment perspective. We found discrepancies concerning cytokines, raising the hypothesis of specific immunological etiological factors. Antibody studies support the existence of a potential autoimmune etiological factor. Infections may also provoke OCD symptoms, and therefore, could be considered as specific etiological factors with specific immunological impairments. Finally, we underline the importance of distinguishing between different etiological factors since some specific treatments already exist in the context of immunological factors for the improvement of classic treatments.

## 1. Introduction

Obsessive–compulsive disorder (OCD) is a major disabling disorder affecting about 2% of the population, and it incurs significant mental health costs [1]. The Diagnostic and Statistical Manual of Mental Disorders, Fifth Edition (DSM-5) defines OCD as comprising two major symptoms: obsessions (i.e., intrusive thoughts or mental images) and compulsions (i.e., repetitive movements or mental acts produced by the patient in response to obsessional thoughts, in order to decrease anxiety) [2].

Several hypotheses exist regarding the physiological basis of OCD with dysfunction of brain circuits involving the limbic cortex and basal ganglia being at the core of the disorder [1,3]. Indeed, several imaging studies found hyperactivity of the orbito-frontal cortex and anterior cingulate cortex [4], and effective treatments for severe forms of OCD act directly on these circuits [5,6]. Some authors proposed hypotheses involving dysfunction of microcircuits within these limbic loops [7]. However, hypotheses constructed to explain the underlying pathology of the disorders make no reference as to the origin of the dysfunctions.

An underlying genetic process could play an important role in the etiology of dysfunctional circuitry. Several genes were found through genomic association studies [1,8]. Among the genes implicated in OCD, dopamine-, glutamate-, or serotonin-related genes are the most studied [1,3], although they are not the only ones to be involved in OCD. A recent study aimed to identify rare de novo mutations based on exomes from 20 OCD parent–child trios patients [9]. Protein mutations were found in developmental and immunological pathways, such as transforming growth factor beta (TGFβ) or complements. These results differ from the usual neurotransmitter gene mutations [9], and they provide arguments for immunological factors in OCD etiology. Furthermore, another study found a significant enrichment of the human leukocyte antigen/antigen D-related 4 (HLA-DR4) serotype allele in OCD patients [10]. According to the results of these genetic studies, abnormalities in immunological mechanisms could lead to OCD, and some specific mechanisms such as microglial dysfunction [11] or autoimmune processes [12] were hypothesized. Furthermore, not only can genes disrupt the immune system through mutations, but the environment can also influence it, through infections for example, subsequently leading to OCD, even with no genetic predisposition [13]. This question of a possible infectious etiology was also suspected in other psychiatric disorders [14].

As many as 25–40% of OCD patients are resistant to classical therapies, such as serotonin recapture inhibitors and cognitive behavioral therapy, and remain so despite advances in OCD treatment such as deep brain stimulation [15,16,17]. Any progress toward a better understanding of the biological basis could provide some solutions for resistant OCD patients. Hence, if a specific cluster of OCD patients with immunological dysfunctions could be determined, some specific treatment could emerge for them and resolve the enduring resistance problem for a minority of patients [18]. The aim of this review was, thus, to summarize the existing immunological data in OCD to show a possible immunological etiological factor in OCD that could be distinct from other factors (e.g., purely genetic OCD), and then, to raise the possibility of a more personalized and effective treatment.

## 2. Method

Our review used the PubMed database. We selected clinical human papers (English language) relevant to the human immunological field concerning OCD. The relevance of an article was based on the abstract published in PubMed. We did not restrain the period of search and reviewed all PubMed results.

Our exclusion criteria were as follows: case report format, small descriptive case series format, commentary format, review format, animal experiments, neurocognitive studies, control group with other psychiatric conditions, absence of OCD data (for example, studies looking for pediatric autoimmune neuropsychological disorders associated with streptococcal infection (PANDAS) etiology only in Tourette’s patients were excluded). Furthermore, therapeutic trials were not selected when these were not targeted by the search terms. For example, when looking for cytokine impairment in OCD through “cytokines AND (OCD OR “obsessive compulsive disorder”)” search terms, some therapeutic trials found were not selected.

When the same paper was found with different search terms, we specify this fact in the tables below. For example, when an article was found with both “cytokines AND (OCD OR “obsessive compulsive disorder”)” and “antibody AND (OCD OR “obsessive compulsive disorder”)” search terms, we detailed the article only in the first table section (here, the “cytokines AND (OCD OR “obsessive compulsive disorder”)” table section); in the second table section (here, the “antibody AND (OCD OR “obsessive compulsive disorder”)” table section), we only mentioned the article and directed the reader to the first table section for details.

The following terms were reviewed:

cytokines AND (OCD OR “obsessive-compulsive disorder”) (67 papers, 22 included); antibody AND (OCD OR “obsessive-compulsive disorder”) (163 papers, 33 included); anti-brain antibody and (OCD OR “obsessive-compulsive disorder”) (6 papers, 3 included); ABGA AND (OCD OR “obsessive-compulsive disorder”) (7 papers, 2 included); “white blood cells” AND (OCD OR “obsessive-compulsive disorder”) (1 paper, 1 included); lymphocyte AND (“obsessive-compulsive disorder” OR OCD) (40 papers, 15 included); monocytes AND (“obsessive-compulsive disorder” OR OCD) (6 papers, 4 included); “NK cells” AND (“obsessive-compulsive disorder” OR OCD) (3 papers, 2 included); infection AND (OCD OR “obsessive-compulsive disorder”) (243 papers, 9 included); Lyme AND (OCD OR “obsessive-compulsive disorder”) (5 papers, 1 included); streptococcus AND (OCD OR “obsessive-compulsive disorder”) (137 papers, 22 included); toxoplasma (OCD OR “obsessive-compulsive disorder”) (8 papers, 4 included); (PANDAS OR PANS) AND treatment AND (OCD OR “obsessive-compulsive disorder” OR tic OR Tourette) (111 papers, 21 included); NSAID and (OCD OR “obsessive–compulsive disorder”) (14 articles, 4 included); “anti-inflammatory” and (OCD OR “obsessive–compulsive disorder”) (21 papers, 5 included); minocycline and (OCD OR “obsessive–compulsive disorder”) (6 papers, 2 included); N-acetylcysteine and (OCD OR “obsessive–compulsive disorder”) (27 papers, 5 included).

The aim of the article was to describe and discuss the potential role of immunological factors in OCD etiology. Hence, even if the method was a systematic one, we wrote our article as a qualitative review to make it easier to read and understand. However, the articles cited in the text are referenced in the tables below, and, when contradictions occurred between articles, this is mentioned and reviewed qualitatively in the text.

## 3. Immunological Changes in OCD

### 3.1. Cytokines

Cytokines are molecules that allow communication between immune cells, or between immune cells and non-immune cells [19]. Studying cytokines can help us understand the mechanism and pathways of potential immunological disruption in OCD. The first studies on cytokine variations in OCD patients included very few patients and were negative except for a positive and significant correlation between IL-6 (interleukin-6) or soluble IL-6 receptor plasma levels and severity of compulsive behaviors [20,21] (Table 1). Further studies [22,23,24,25,26,27,28,29,30] were carried out enabling a meta-analysis [31] (Table 1), which found decreased IL-1β levels and decreased TNFα (tumor necrosis factor α) levels in non-depressed OCD patients (but not in OCD patients with possible comorbid depression), and increased IL-6 levels in adult medication-free OCD patients (but not in OCD children with possible medication use) compared to controls. More recently, discrepancies were found with this previous meta-analysis concerning TNF-α with increased levels in OCD patients [32,33,34] (Table 1). Despite these discrepancies, the increased IL-6 levels seem a consistent result as they were replicated in a recent study [34,35] (Table 1).

The fact that IL-6 levels are higher in autoimmune diseases (e.g., rheumatoid arthritis) [19] raises the hypothesis that increased IL-6 levels in OCD could favor the existence of an autoimmune etiological factor. Furthermore, tocilizumab, an IL-6 receptor-neutralizing antibody, appears as a putative treatment in some cases of OCD as this molecule was always found effective in some autoimmune disorders where IL-6 is involved [19]. We could, thus, hypothesize that tocilizumab or other specific immunological treatments could help some specific OCD patients with a possible immunological etiological factor.

It is known also that both IL-6 and TNFα can be involved in asthma pathophysiology and allergic diseases [42], and allergic diseases would appear to be more frequent in OCD patients [43], giving weight to elevated IL-6 and TNFα levels in some OCD cases.

TNFα, IL-1β, and IL-6 are inflammatory cytokines (for a very complete review, see Reference [42]). TNFα is produced by a wide range of cells including T- or B-cells and monocytes (including microglia), and it targets all nucleated cells. TNFα has a complex role, being both pro-inflammatory and immunosuppressive. In the brain, TNFα could be involved in synapses scaling with high levels of TNFα favoring LTP (long term potentiation) and low levels of TNFα favoring LTD (long term depression) [44,45]. Progranulin mutations were found to be associated with hyperexcitability of nucleus accumbens spiny neurons in mice, in line with hyperactivity of cortico-striatal loops in OCD [1], and elevated TNFα levels and hyperactivation of microglia [46]. With the progranulin gene restored, OCD-like behavior disappeared in mice [46]. Frontoparietal dementia patients showing mutations of progranulin presented OCD [46]. 

IL-1β is also produced by microglia and targets T-cells or endothelial and epithelial cells [42]. 

IL-6 is produced by both astrocytes and microglia, and IL-6 exposure could increase synaptic activity (for an excellent review on IL-6 central nervous system (CNS) effects, see Reference [47]).

In summary, the literature on cytokines involved in OCD is difficult to interpret as contradictory results are often found. These discrepancies could be due to the heterogeneity of patients studied, e.g., children vs. adults, and they emphasize the importance of considering the immunological status of recruited patients. Thus, differences in symptoms, and their development or response to treatment between OCD patients with and without modified IL-6 and TNFα levels would suggest the possibility of a specific immunological OCD cluster.

### 3.2. Antibodies

Most studies concerning antibodies in OCD concern pediatric autoimmune neuropsychological disorders associated with streptococcal (PANDAS) infections.

Studies found that a subset of patients suffering from OCD showed high levels of anti-basal ganglia antibodies (ABGAs) and anti-streptolysin O antibodies (ASO) in the blood or corticospinal fluid (CSF) [48,49,50,51] (Table 2). These studies strongly support the existence of an autoimmune etiological factor in OCD. However, discrepancies still exist: ABGAs were found in OCD patients (and not in controls), but not in all OCD patients [49] (Table 2).

Furthermore, it was shown that patients suffering from rheumatic fever—a disorder linked with PANDAS—show a higher proportion of a specific B-lymphocyte alloantigen detected with monoclonal antibodies D8/D17 [82,83]. Consequently, some authors tried determining whether the monoclonal antibody D8/D17 could also be used as an OCD marker. The D8/D17 mean value was shown to be higher in OCD or Tourette’s child populations vs. control subjects [80], and, as no difference was found between Tourette syndrome and OCD patients in D8/D17 values, it was hypothesized that it could be a marker for OCD in children [80]. Although these results were replicated [74,77], there is still debate surrounding this topic [71,73,81,84] (Table 2).

However, this higher proportion of B-lymphocyte alloantigen detected with monoclonal antibody D8/D17 found in some OCD patients could be a promising way of classifying patients in specific subgroups of OCD (patients with a high D8/D17 value vs. OCD patients without) and could, thus, be a promising line for proposing more specific treatments for these particular patients.

### 3.3. White Blood Cells

As with cytokines, very contrasting results were found for white blood cells. Some studies found a higher number of CD8+ (cluster od differentiation) lymphocytes and a lower number of CD4+ lymphocytes in OCD patients [76], whereas others did not [85,86] (Table 3). Furthermore, other studies concerning different white blood cells (monocytes or NK cells (natural killer cells)) also found different results [28,35,87] (Table 3). While cytokines are studied intensively in the context of OCD, more studies will need to be done specifically on white-blood-cell counts and activity.

## 4. Infections and OCD

Here, we describe two of these infectious etiological factors (Table 4), and we discuss some possible mechanisms via which these infectious agents could lead to OCD, and hence, why these infection contexts could be considered as specific OCD subtypes.

### 4.1. Streptococcal Infection

*Streptococcus pyogenes* is a bacterial group that can lead to several pathologies such as pharyngitis, scarlet fever, or erysipelas [106]. Among these diseases, rheumatic fever is the one we were interested in. This disease is characterized by elevated ASO (anti-streptolysin O) or anti-DNAse B antibody levels [107], and it affects the heart, skin, bone joints, and CNS [108,109]. The Jones criteria are usually used to make the diagnosis. They consist of carditis, arthritis, chorea, erythema marginatum, and subcutaneous nodules with evidence of *S. pyogenes* infection [108,109]. *S. pyogenes* can, thus, affect the nervous system through choreic movements. This chorea, called Sydenham’s chorea [110,111], is characterized by involuntary movements which are irregular, rapid, and transient, and which are typically manifested in the extremities [111,112]. Sydenham’s chorea is characterized by antibodies found in the basal ganglia that react with *N*-acetyl-beta-d-glucosamine of *S. pyogenes*, and with lysoganglioside and tubulin of the brain [111,113]. This cross-reaction is made possible by a mimicry process [111]. Furthermore, it was shown recently by Cox and colleagues that these antibodies could react with the D2-receptor (D2R) complex, which could be causal in Sydenham’s chorea, as risperidone reverses this movement disorder [114]. In summary, antibodies that originally target *S. pyogenes* may also attack the patient’s brain.

Since the basal ganglia (where Sydenham’s chorea antibodies are found) appear to be a key region in OCD neurobiology [1,3], one could imagine that antibodies against basal ganglia (which seems the case in PANDAS [68]) could impair their functioning and lead to OCD symptoms in some conditions. In this context, it is notable that OCD may be associated with Sydenham’s chorea [115]. In addition to this association, the concept of PANDAS (pediatric autoimmune neuropsychological disorder associated with streptococcal infection) was originally defined by Swedo et al. in 1998 [105] as follows: presence of OCD or tic disorder, symptom onset between the age of three and puberty, exacerbation of symptoms associated with streptococcal infection, and presence of neurological abnormalities during symptom exacerbation, but in the absence of frank chorea which would suggest Sydenham’s chorea [105]. This original description of PANDAS was modified in 2012 by Swedo et al. to become PANS (pediatric acute-onset neuropsychiatric syndrome, with abrupt onset of OCD or severely restricted food intake and presence of additional neuropsychiatric symptoms such as anxiety, emotional liability, etc.) [13]. PANDAS and PANS could, thus, constitute a specific OCD subgroup for which the underlying physiological mechanism could be the same, that is to say, an autoantibody against basal ganglia neurons [110,113].

However, no D2R antibodies were found in PANDAS patients [12]. Cox et al. recently studied patients with tic disorders or OCD or both, and with a history of streptococcal infection [58]. They found that these patients as a whole presented elevated levels of anti D1-receptor (D1R) antibodies in the serum (with elevated anti-lysoganglioside antibodies) compared to controls. As in Sydenham’s chorea, anti-lysoganglioside antibodies seem to be involved [58,65] (Table 4).

Recently, antibodies in children suffering from PANDAS were found to bind more to cholinergic interneurons of mice than control antibodies when mice were infused with patient and control serum in their striatum [116]. Taken together, these results raise the question of the proportions of dopamine receptor subtypes and the role of cholinergic interneurons in OCD and more particularly in PANDAS, which is a good example of the multiple etiologies of OCD. It is one of the rare clearly identified etiological factors of OCD. About 5% of pediatric OCD patients meet the criteria for PANDAS (or PANS) [117] and it is important to distinguish this etiology from others in OCD patients. Indeed, the prognosis of PANDAS seems relatively good, as Leon et al. found that 88% of children originally suffering from PANDAS with moderate-to-severe OCD presented no OCD symptoms (55%) or only subclinical symptoms (33%) after approximately three years of follow-up [118] (Table 5). This result of a good prognosis is confirmed by Murphy et al. [61], but not by Frankovich et al. [57] (Table 5). By contrast, 48% of OCD patients were found to be still symptomatic after 30 years [119]. However, as this study began in 1954, and PANDAS patients, which represent about 5% of OCD patients, were first described in 1998 [105,117], one may hypothesize that PANDAS and non-PANDAS OCD patients were pooled together. Furthermore, treatment of PANDAS (described below) is not identical to OCD treatment. Prophylactic antibiotics or antibiotic treatment, anti-inflammatory treatment, and immunoglobulin, indicated in PANDAS treatment, are not prescribed in “idiopathic OCD” [118,120,121,122,123,124,125]. Therefore, it would be important to recognize and adequately treat PANDAS within a personalized medical setting.

### 4.2. Toxoplasma gondii

*Toxoplasma gondii* is an intracellular parasite that is linked to several psychiatric disorders including schizophrenia and bipolar disorder [144,145]. *T. gondii* is also linked to OCD [14,54,63,145] (Table 4). A recent study found that the presence of anti-*Toxoplasma gondii* IgG (immunoglobulin G) in serum was more frequent in OCD patients than in controls (the odds ratio (OR) was 4.84 (confidence intervals = 1.78–13.12) in favor of OCD) [52] (Table 4). Furthermore, in a 1991 study, Strittmatter and colleagues showed that the CNS areas most affected by *T. gondii* were the cerebral hemispheres (91%) and the basal ganglia (78%) which are implicated in OCD neurobiology [146]. There are several hypotheses regarding how *T. gondii* reaches the CNS (for a review, see the article by Ueno et al. [147]). Among these, the monocyte hypothesis is of particular interest. Indeed, the fact that *T. gondii* is found in the brain CD11b+ monocytes, which can be microglial cells (the resident monocytes of the brain) [148], suggests that *T. gondii* can invade monocytes in the peripheral blood supply and then reach the brain. Once in the brain and in microglia, these monocytes become activated and show increased migratory activity [149].

*T. gondii* infection leads to IFN-γ (interferon) production, and then, to the induction of IDO (indolamine-2,3-dioxygenase), mainly produced by microglia and one of the main enzymes of kynurenine pathway [150,151,152,153,154]. This induction of IDO by *T. gondii* occurs firstly in parallel with the *T. gondii*-induced microglia activation and can secondly lead to a tryptophan depletion (since the kynurenine pathway is part of tryptophan catabolism) [149,150,155]. As tryptophan is the essential amino acid for serotonin synthesis, tryptophan depletion could interfere with OCD physiological pathways since OCD symptoms are improved with specific serotonin reuptake inhibitors (SSRIs) [156,157,158]. Nonetheless, we have to keep in mind that this causal tryptophan depletion hypothesis is still a matter of debate in MDD (major depressive disorder); thus, the putative *T. gondii* role in OCD is unclear [159].

We could also hypothesize a *T. gondii* action at the level of striatal dopamine receptors. *T. gondii* contains genes coding for tyrosine hydroxylase, and it was shown that *T. gondii* increases the dopamine release [160]; therefore, *T. gondii* could lead to OCD through dopamine release and its action on striatal D1 receptors, and then, via the activation of the direct pathway (associated with the D1 receptor [161]). Nonetheless, this hypothesis is highly speculative since interactions among D1 receptors, D2 receptors (between direct and indirect pathway), and serotonin receptors are complex, and D1 receptor downregulation could be a consequence of a D1 receptor hyper stimulation, thus leading to an inhibition of the direct pathway.

Finally, there is a neurotoxic hypothesis, via the direct neurotoxic role of quinolinic acid (produced by the kynurenine pathway) and IFN-γ which acts as a neurotoxic agent through its action on the kynurenine pathway [162]. Therefore, one could speculate that *T. gondii* is neurotoxic for the striatal microcircuit, and thus, contributes to the occurrence of OCD symptoms.

According to these different hypotheses on the role of *T. gondii* in the genesis of OCD, some innovative treatment options might be suggested such as the use of IDO inhibitors used for some cancer treatments [163], which were already tested in some animal models of schizophrenia where such treatment seems to protect the striatum from the negative effects of kynurenine pathway activation [164].

## 5. Alternative Treatments for OCD

The above distinct etiological factors in OCD could be taken into account to develop specific treatments. Tricyclic or SSRI antidepressants are the usual treatment for OCD [165]. For refractory and severe OCD, deep brain stimulation can also be used [6]. Other treatments were also developed for specific etiological factors.

### 5.1. Specific Treatment in the PANS/PANDAS Context

Several specific treatments were studied for PANS/PANDAS patients. Intravenous immunoglobulin (IVIG) could be an effective treatment [120,124,166] (Table 5); however, its effectiveness remains to be confirmed. Another treatment procedure studied was apheresis. Two studies found this treatment to be effective [128,166] (Table 5), but they suffered from limitations (absence of a control group or a limited number of patients studied), which meant no definitive conclusion could be drawn on its effectiveness in PANDAS. The effects of antibiotic treatment in PANDAS were also studied. Four studies without control groups found that antibiotics could be effective [118,120,129,133] (Table 5), although a study comparing azithromycin vs. placebo as a treatment for PANS over four weeks failed to find an effect of azithromycin on OCD symptoms as measured with the Children’s Yale–Brown Obsessive Compulsive Scale (CY-BOCS) [123] (Table 5). However, if streptococcal infection is still present during acute episodes of PANDAS, antibiotics are considered as the best treatment [167,168]. Finally, corticosteroid and nonsteroidal anti-inflammatory drugs (NSAIDs) do appear to be effective [120,121,122,126] (Table 5).

Hence, several alternative specific treatments to PANDAS/PANS were studied. However, even if some of these proposed treatments seem promising, robust clinical evidence is still lacking to allow us to reach a definitive conclusion [169].

### 5.2. Specific Treatment in the “Classical” OCD Context

In addition to PANS/PANDAS, immunological treatments were also tested in “classical” OCD, that is to say, with no clearly identified etiological factor. NSAIDs show contrasting results [135,136,170] (Table 5). However, in the general context of OCD with no specific etiological factors, anti-inflammatory treatment seems to have a place in the treatment strategy, which could be more precisely defined if OCD etiologies were better known. Minocycline, a specific antibiotic, is particularly interesting because of its action on microglia (see below). Minocycline was studied as a potential new pharmacological treatment for OCD, and the results were mixed: one study found that minocycline could be a good adjunctive treatment to classical OCD treatment with SSRIs [137], but another one did not find this result [138] (Table 5). Another antibiotic, cefdinir, was studied, but it showed no effectiveness on the CY-BOCS scores when compared to a placebo [171]. Finally, N-acetylcysteine (NAC), an antioxidant which has a neuroprotective role against oxidative stress, produced divergent results [139,140,141,142,143] (Table 5).

These different studies on immunological treatment in the PANDAS/PANS contexts or otherwise indicate that some specific treatments for different aspects of immunity could have a place in OCD treatment.

## 6. Conclusions: Future Lines of Research for Etiological Immune Response Factors

### 6.1. Animal Models

The rodent animal model is widely used in anxiety disorder studies. Rodents present many behavioral signs of anxiety in various contexts. Among these behaviors, grooming was considered by some authors as a compulsive-like behavior, due to its repetitive and sequential organization [172]. Hence, several animal models that were created by mutating genes of interest (e.g., *Sapap3* mutant-mice [173]) were considered as animal models for OCD because of their excessive grooming behavior among other parameters. *Hox* genes are homeotic genes [174], which are responsible for the anterior posterior segmentation of the organism. They are also involved in the formation of the hematopoietic system, and *Hoxb8* is especially involved in the differentiation of myeloid progenitor cells, one source of microglial cells [175]. It is, therefore, of note that, firstly, *Hoxb8* mutant-mice show exacerbated grooming behavior since, in the brain, microglia are the only cells linked to *Hoxb8*, and secondly, that grooming could be reversed after normal bone marrow transplantation which allows *Hoxb8*-derived microglia to migrate to the brain [175]. These data show a direct involvement of microglia, which is an immunological component in compulsive-like behaviors. Therefore, microglia (see below) could be a promising future line of research to better understand OCD and the role of immunology in a specific OCD cluster.

Rats exposed to *Streptococcus* antigens show more grooming behavior than control rats [176] and offer a model within which to investigate the link between OCD neurobiology and PANDAS. Indeed, grooming behavior was found to be reversible with serotonin re-uptake inhibitors, and IgG was found in key brain areas of OCD rats (i.e., striatum, thalamus, and frontal cortex), and glutamate and dopamine levels were also found to be modified [176]. Furthermore, the amelioration of some of the previous abnormalities found in this model with an antibiotic treatment [177] is in line with other results in humans [123].

### 6.2. Microglia

Tourette syndrome is a condition close to the OCD spectrum [178,179], and is well known for showing interneuron loss [180]. Interestingly, a high level of CD45, which is a marker of activated microglia, was found in Tourette’s post-mortem basal ganglia [181,182]. An elevated expression of CCL2 (chemokine ligand 2), which is a chemokine that activates microglia, was also found in these brains [181,182], raising the question of whether the interneuron loss in Tourette syndrome is linked in some manner to microglia activation. Indeed, among the functions of microglia are synapse elimination and phagocytosis [183]. There is convergent evidence in OCD, as a recent study found microglia activation in an OCD brain employing a PET (positron emission tomography) protocol [184], which shows a potential role for inflammation and microglia in OCD neurobiology [11]. These results are consistent with the cytokine levels found in OCD patients, as activated microglia produce IL-6, IL-1β, and TNF-α [11,181]. This could explain the effectiveness of minocycline, which reduces microglia activation [185], as another OCD treatment in Reference [137], and highlights the importance for precision medicine to consider immunological etiological factors.

### 6.3. The Importance of the Attempt to Identify Different OCD Etiologies

As we can see (Table 6), it is likely that there are multiple etiological factors in OCD. Genetic and environmental factors clearly play a role in the emergence of OCD. Some genetic studies indicate the involvement of immune response genes in the physiological basis of OCD [9,10,186]. The environment could play a role through epigenetic processes [187], and could also have a more direct influence on brain function through immunological processes, as is the case with PANDAS. Furthermore, it was shown that stress may directly impact some immunological parameters [188], raising the putative role of psychological stressors through immunological responses in OCD emergence.

Future research should focus on these etiological factors (genetic, immunological, etc.) in order to elucidate the biological bases of OCD, and to develop prevention tools and better treatments [189], paving the way to precision individualized therapies [190] for the benefit of patients. The identification of more specific biological clusters in OCD is essential in order to advance our knowledge and treatment of OCD.

## Figures and Tables

**Table 1 brainsci-08-00149-t001:** Cytokine studies.

Cytokines AND (OCD OR “Obsessive Compulsive Disorder”)			
Authors, Date	Subjects	Main Results	Significance
Jiang C. et al. (2018) [36] Meta-analysis	435 cases 1073 controls	TNF-α polymorphisms	
		-> G vs. A model: OR = 1.01; 95% CIs = 0.37–2.77;	*p* = 0.981
		-> GG vs. AA + AG model: OR = 0.93; 95% CIs = 0.37–2.37;	*p* = 0.879
		-> GG + AG vs. AA model: OR = 0.22; 95% CIs = 0.06–0.73;	*p* = 0.014
		-> GG vs. AA model: OR = 0.21; 95% CIs = 0.06–0.71;	*p* = 0.12
		-> AG + AA model: OR = 0.29; 95% CIs = 0.07–1.16;	*p* = 0.081
		-> GG + AA vs. AG model: OR = 1.17; 95%CIs = 0.55–2.51;	*p* = 0.683
Colak Sivri R. et al. (2018) [32]	44 OCD patients 40 controls	-> OCD log-TNF-α > controls log-TNF-α	*p* < 0.001
		-> OCD log-IL-12 < controls log-IL-12	*p* = 0.014
		No difference concerning BDNF, TFG-β (tendency of increased level in OCD patients), IL-1β (tendency of decreased level in OCD patients), IL-17, sTNFR1, sTNFR2, CCL3, CCL24 (tendency of increased level in OCD patients), CCL8	
Rodriguez N. et al. (2017) [35]	102 OCD patients 47 controls	-> Monocytes percentage of OCD patients > controls	
		-> CD16+ monocytes percentage of OCD patients > controls	
		After LPS stimulation	
		-> OCD-patients IL-1β > controls IL-1β	*p* = 0.005
		-> OCD-patients IL-6 > controls IL-6	*p* = 0.004
		-> OCD-patients GM-CSF > controls GM-CSF	*p* = 0.049
		-> OCD-patients TNF-α > controls TNF-α	*p* = 0.041
		-> OCD-patients IL-8 > controls IL-8	*p* = 0.013
Simsek S. et al. (2016) [33]	34 OCD patients 34 controls	-> OCD patients IL-17α > controls IL-17α	*p* = 0.03
		-> OCD patients TNF-α > controls TNF-α	*p* = 0.01
		-> OCD patients IL-2 > controls IL-2	*p* = 0.02
		No difference for IFNγ, IL-10, IL-6, IL-4 (tendency of increased level in OCD patients)	
Rao NP. et al. (2015) [34]	20 OCD patients 20 controls	-> OCD patients IL-2 > controls IL-2	*p* = 0.005
		-> OCD patients IL-4 > controls IL-4	*p* = 0.007
		-> OCD patients IL-6 > controls IL-6	*p* = 0.002
		-> OCD patients IL-10 > controls IL-10	*p* = 0.006
		-> OCD patients TNF-α > controls TNF-α	*p* = 0.005
		No difference concerning IFN-γ	
Uguz F. et al. (2014) [37]	7 OCD patients 30 controls	-> cord blood TNF-α of new born infants of women with OCD > cord blood TNF-α of new born infants of control women	*p* = 0.036
Bo Y. et al. (2013) [38]	241 OCD patients 444 controls	IL-1β-511C/T polymorphism	
		No difference between OCD patients and controls	
Zhang X. et al. (2012) [39]	200 OCD patients 294 controls	MCP-1-2518G/A polymorphism	
		No difference between OCD patients and controls	
Liu S. et al. (2012) [40]	187 OCD patients 281 controls	IL-8-251T/A polymorphism	
		No difference	
Gray SM. et al. (2012) [31] Meta-analysis	169 OCD patients 215 controls	-> Decreased IL-1β in OCD patients	*p* < 0.01
		-> Increased IL-6 in adult free-medication OCD patients	*p* = 0.02
		No difference concerning IL-6 in OCD children	
		-> Decreased TNF-α in OCD patients without depression	*p* < 0.001
		No difference in TNF-α when depressed patients are considered	
Cappi C. et al. (2012) [22]	183 OCD patients 249 controls	TNF-α A/G polymorphism	
		-> Association of allele A with OCD (χ^2^, rs361525)	*p* = 0.007
Fontenelle LF. et al. (2012) [23]	40 OCD patients 40 controls	-> OCD patients CCL3 > controls CCL3	*p* = 0.03
		-> OCD patients CXCL8 > controls CXCL8	*p* < 0.001
		-> OCD patients sTNFR1 > controls sTNFR1	*p* < 0.001
		-> OCD patients sTNFR2 > controls sTNFR2	*p* < 0.01
		No difference between OCD and controls concerning CCL2, CCL11, CCL24 (tendency of increased level in OCD patients), CXCL9, CXCL10 (tendency of decreased level in OCD patients), IL-1ra, TNF-α.	
Fluitman SB et al. (2010) [24]	10 OCD patients 10 controls	During disgust exposure:	
		-> LPS-stimulated TNF-α in OCD patients decreased after disgust exposure	*p* = 0.07
		LPS-stimulated TNF-α in controls not changed after disgust exposure	
		-> LPS-stimulated IL-6 in OCD patients decreased after disgust exposure	*p* = 0.040
		LPS-stimulated IL-6 in control not changed after disgust exposure	
Fluitman S. et al. (2010) [25]	26 OCD patients 52 controls	-> OCD patients LPS-stimulated IL-6 < control LPS-stimulated IL-6	*p* = 0.016
		No difference concerning LPS-stimulated IL-8 and TNF-α	
Hounie AG et al. (2008) [26]	111 OCD patients 250 controls	TNF-α-A/G polymorphism	*p* = 0.0005 and
		-> Association of the A allele with OCD for 238 G/A and 308 G/A (χ^2^)	0.007 respectively
Konuk N. et al. (2007) [27]	31 OCD patients 31 controls	-> OCD patients TNF-α > control TNF-α	*p* < 0.001
		-> OCD patients IL-6 > control IL-6	*p* < 0.001
Denys D. et al. (2004) [28]	50 OCD patients 25 controls	-> OCD patients LPS stimulated IL-6 > control LPS stimulated IL-6	*p* = 0.004
		-> OCD patients LPS stimulated TNF-α > control LPS stimulated TNF-α	*p* < 0.001
		-> decreased NK cells activity in OCD patients	*p* = 0.002
		No difference concerning LPS-stimulated IL-10	
Carpenter LL. et al. (2002) [41]	26 OCD patients 26 controls	No difference concerning CSF IL-6 level.	
Monteleone P. et al. (1998) [29]	14 OCD patients 14 controls	-> OCD patients TNF-α < control TNF-α	*p* = 0.001
		No difference concerning IL-6 and IL-1β	
Brambilla F. et al. (1997) [30]	27 OCD patients 27 controls	-> OCD patients IL-1β < control IL-1β	*p* = 0.0004
		-> OCD patients TNF-α < control TNF-α	*p* = 0.0004
Weizman R. et al. (1996) [20]	11 OCD patients 11 controls	No difference concerning IL-1β, IL-2, and IL-3-LA production between OCD patients and controls	
Maes M. et al. (1994) [21]	19 OCD patients 19 controls	No difference concerning IL-1β, IL-6, sIL-2R, sIL-6R	

A/G: adenine/guanine; BDNF: brain-derived neurotrophic factor; C/T: cytosine/thymine; CCL: chemokine ligand; CIs = confidence interval; CSF: cerebrospinal fluid; CXCL: chemokine (C-X-C motif) ligand; G/A: guanine/adenine; GM-CSF: granulocyte-macrophage colony stimulating factor; IFN: interferon; IL: interleukin; IL-1ra: interleukin 1 receptor antagonist; LA: like activity; LPS = lipopolysaccharide; MCP: monocyte chemoattractant protein; NK: natural killer; OCD: obsessive-compulsive disorder; OR = odds ratio; sIL-2R: soluble interleukine-2 receptor; sTNFR: soluble tumor necrosis factor receptor; T/A: thymine/adenine; TNF: tumor necrosis factor.

**Table 2 brainsci-08-00149-t002:** Autoimmunity and OCD.

Antibody AND (OCD OR “Obsessive Compulsive Disorder”)			
Authors, Date	Subjects	Main Results	Significance
Akaltun I. et al. (2018) [52]	60 OCD 60 controls	-> Toxoplasma IgG levels related to OCD status	*p* = 0.001
		-> IgG positivity individuals: increased risk of OCD: OR = 4.84, 95% CIs = 1.78–13.12	*p* = 0.002
Mataix-Cols D. et al. (2017) [53]	30082 OCD 472874 patients	-> Augmentation of the risk to develop autoimmune disease: OR = 1.43; 95% CIs = 1.37–1.49	*p* < 0.01
Flegr J. et al. (2017) [54]	281 men and 831 women not infected 65 men and 350 women infected with toxoplasma	-> Association between toxoplasma infection and OCD: OR = 2.27, 95% CIs = 1.01–5.09	*p* = 0.047
Sutterland AL. et al. (2015) [14] Meta-analysis	No information but 2 studies included	-> Association between OCD status and toxoplasma infection: OR = 3.4; 95% CIs = 1.73–6.68	*p* = 0.0004
Nicolini H. et al. (2015) [55]	37 PANDAS/OCD or tics patients 12 controls	-> OCD patients anti-enolase > controls anti-enolase	*p* = 0.035
		-> OCD patients anti-streptococcal proteins > controls anti-streptococcal proteins	*p* = 0.05
		No differences concerning anti-neural antibodies.	
Singer HS. et al. (2015) [56]	8 PANDAS/OCD or tics patients 70 controls	No association between clinical exacerbation and anti-tubulin, anti-lysoganglioside GM1, anti D1R, anti D2R titer.	
Frankovich J. et al. (2015) [57]	19 PANS/OCD or eating disorder patients 28 non-PANS but OCD or eating disorder patients	No difference concerning comorbidities (anxiety, mood disorder, irritability, suicidality)	
		No difference concerning Ig levels	
		No difference concerning remitting course, chronic course.	
Cox CJ. et al. (2015) [58]	311 PANDAS/OCD or tics patients 16 controls	-> PANDAS patients anti-D1R patients > controls anti-D1R	*p* < 0.0001
		-> PANDAS patients anti-lysoganglioside > controls anti-lysoganglioside	*p* = 0.0001
Ebrahimi Taj F. et al. (2015) [59]	76 OCD/ADHD patients 39 controls	-> OCD/ADHD patients anti-streptolysin O > controls anti-streptolysin O	*p* < 0.0001
		-> OCD/ADHD patients anti-streptokinase > controls anti-streptokinase	*p* < 0.0001
		-> OCD/ADHD patients anti-DNase B > controls anti-DNase B	*p* < 0.0001
Murphy TK. et al. (2015) [60]	43 PANS/OCD patients	infectious triggers: 58% of GAS, 12% of mycoplasma pneumoniae, 37 of upper respiratory infection, 2% of Lyme	
		No differences between patients with tics and without tics concerning anti-DNase B, ASO, Mycoplasma IgM/IgG, Lyme screen, age of onset, CY-BOCS score, Y-GTSS score	
Murphy TK. et al. (2012) [61]	41 PANDAS/OCD or tic patients 68 non-PANDAS but OCD or tic patients	-> PANDAS patients remissions > non-PANDAS patients remissions	*p* < 0.05
		-> PANDAS patients dramatic onset > non-PANDAS patients dramatic onset	*p* < 0.05
		-> PANDAS patients ASO/anti-DNase > non-PANDAS patients ASO/anti-DNaseB	*p* < 0.0001
		-> remission in PANDAS patients after antibiotic treatment > remission in non-PANDAS after antibiotic treatment	*p* < 0.01
Leckman JF et al. (2011) [62]	31 PANDAS/OCD or tic patients 53 non-PANDAS/OCD or tic patients	No association between clinical exacerbation and new GAS infection.	
Miman O. et al. (2010) [63]	42 OCD patients 100 controls	-> OCD patients anti-toxoplasma IgG > controls anti-toxoplasma IgG	*p* < 0.01
Bhattacharyya S. et al. (2009) [48]	23 OCD patients 23 controls	-> more CSF anti-brain antibody binding to basal ganglia and thalamus for OCD patients than for patients	*p* < 0.05
		-> More CSF glutamate and glycine in OCD patients than in controls	*p* < 0.001
Gause C. et al. (2009) [64]	13 OCD only patients 20 PANDAS/OCD patients 23 PANDAS/tic patients 29 controls	No difference concerning ASO titers	
		No difference concerning serum IgG	
		-> More anti-neural antibodies PANDAS/OCD than in other groups	*p* < 0.009
Morer A. et al. (2008) [49]	32 OCD patients 19 controls	No anti-basal ganglia antibody detected by immunohistochemistry	
		-> Anti-basal ganglia antibodies in OCD patients and no in control detected by immunoscreening	
		No difference concerning ASO titers	
Kirvan CA. et al. (2006) [65]	16 PANDAS/OCD or tic patients 25 non-	-> lysoganglioside GM1 concentration required to inhibit binding PANDAS sera to GlcNAc (an epitope of GAS carbohydrate) < lysoganglioside GM1 concentration required to inhibit binding non-PANDAS sera to GlcNAc (an epitope of GAS carbohydrate)	*p* < 0.05
	PANDAS/OCD or tic or ADHD patients	-> lysoganglioside GM1 = specific inhibitor of PANDAS IgG binding to GlcNAc	
		-> PANDAS sera induced activation of CaM kinase II more than non-PANDAS sera => PANDAS serum responsible for cell signaling	*p* = 0.001
Morer A. et al. (2006) [66]	18 early onset OCD 21 late onset OCD	-> Child OCD ASO titer > adult OCD ASO titer	*p* = 0.031
		No difference for D8/D17	
Singer HS. et al. (2005) [67]	48 PANDAS (OCD or tic status not informed) patients 43 controls	No median ELISA optical density difference concerning serum antibodies	
		No difference concerning reactivity against pyruvate kinase M1, α-enolase, γ-enolase, aldolase C	
Pavone P. et al. (2004) [68]	22 PANDAS (OCD or tic status no informed) patients 22 GAS uncomplicated infected patients	-> PANDAS anti-basal ganglia antibody > GAS patients anti-basal ganglia antibody	*p* < 0.001
		No difference concerning ASO or anti DNase B antibody	
Murphy TK. et al. (2004) [69]	15 OCD or tics patients with large symptom fluctuations 10 OCD or tics patients without large symptom fluctuations	-> positive correlation between OCD severity and ASO titer in patients with large symptom fluctuations	*p* = 0.0130
Luo F. et al. (2004) [70]	47 OCD or tic patients 19 controls	-> OCD or tic patients percentages of D8/D17 positive cells > controls percentages of D8/D17 positive cells	*p* = 0.0029
Inoff-Germain G. et al. (2003) [71]	108 positive children for D8/17 marker 132 negative dor D8/17 marker	No association between D8/17 marker status and OCD or tic status	
Murphy ML. et al. (2002) [72]	12 PANDAS OCD patients	-> abrupt appearance of OCD symptoms	
		-> elevated anti-DNase B titer	
		-> mean age at onset = 7 years	
Eisen JL. et al. (2001) [73]	29 OCD patients 26 controls	No difference in D8/D17 marker positivity	
Murphy TK. et al. (2001) [74]	32 OCD or tic patients 12 controls	-> OCD/tic patients D8/D17 titers > control D8/17 titers	*p* = 0.01
Peterson BS. et al. (2000) [75]	105 tic, OCD or ADHD patients 37 controls	No association between OCD or tic disorder and ASO or anti-DNase B titers	No access to p-values
		-> ASO or anti-DNase B titers positively correlated with putamen or globus pallidus volume in OCD patients	
Marazziti D. et al. (1999) [76]	20 OCD patients 20 controls	-> Increased CD8+ lymphocytes in OCD patients	*p* = 0.002
		-> decreased CD4+ lymphocytes in OCD patients	*p* = 0.003
Chapman F. et al. (1998) [77]	41 OCD or tic patients 31 controls	-> OCD or tic patients D8/D17 positivity > control D8/D17 positivity	*P* < 0.0001
Khanna S. et al. (1997) [78]	76 OCD patients 55 controls	-> OCD patients mumps and HSV-I IgG > control mumps and HSV-I IgG	*p* < 0.05
Khanna S. et al. (1997) [79]	76 OCD patients 30 controls	-> OCD patients measles CSF IgG < control measles CSF IgG	*p* < 0.001
		-> OCD patient herpes CSF IgG > control herpes CSF IgG	*p* < 0.05
Murphy TK. et al. (1997) [80]	31 OCD or tic patients 21 controls	-> OCD patients D8/17 positivity > control D8/17 positivity	*p* < 0.001
		No difference concerning ASO, anti-DNase B and anti-neural antibodies	
Swedo SE. et al. (1997) [81]	27 PANDAS/OCD or tic patients 24 controls	-> PANDAS/OCD or tic patients D8/D17 positivity > control D8/D17 positivity	*p* < 0.0001
Anti-brain antibody and (OCD OR “obsessive–compulsive disorder”)			
Bhattacharyya S. et al. (2009) [48]	Cf. antibody AND (OCD OR “obsessive compulsive disorder”)		
Dale RC. et al. (2005) [50]	50 OCD patients 40 controls with uncomplicated streptococcal infection	-> ABGA level in OCD patients > ABGA level in controls	*p* < 0.005
Pavone P. et al. (2004) [68]	Cf. antibody AND (OCD OR “obsessive compulsive disorder”)		
ABGA and (OCD OR “obsessive–compulsive disorder”)			
Pearlman DM. et al. (2014) [51] Meta-analysis	297 OCD patients 406 controls	-> ABGA seropositivity in OCD patients > ABGA seropositivity in controls	*p* < 0.0001
Dale RC. et al. (2005) [50]	Cf. Anti-brain antibody and (OCD OR “obsessive–compulsive disorder”)		

ABGA = anti-basal ganglia antibody; ADHD = attention deficit/hyperactivity disorder; ASO = anti-streptolysin O; CaM Kinase II = Ca^2+^/calmodulin dependent protein kinase II; CD = cluster of differentiation; CIs = confidence intervals; CSF = cerebrospinal fluid; CY-BOCS = children’s Yale–Brown obsessive-compulsive scale; D1R = dopamine 1 receptor;D2R = dopamine 2 receptor; D_ and D17 = B lymphocyte antigen; DNase = deoxyribonuclease; GAS = group A streptococcus; GM1 = monosialotetrahexosylganglioside 1; HSV = herpes simplex virus; Ig = immunoglobulin; OCD: obsessive-compulsive disorder; OR = odds ratio; PANDAS = pediatric autoimmune neuropsychological disorders associated with streptococcal infection; PANS = pediatric acute-onset neuropsychiatric syndrome; Y-GTSS = Yale global tic severity scale.

**Table 3 brainsci-08-00149-t003:** White blood cells and OCD.

”White Blood Cells” OR “Total Blood Count” AND (OCD OR “Obsessive Compulsive Disorder”)			
Authors, Date	Subjects	Main Results	Significance
Atmaca M. et al. (2011) [85]	30 OCD patients 30 controls	-> OCD patients neutrophils < control neutrophils	*p* < 0.05
lymphocyte AND (“obsessive compulsive disorder” OR OCD)			
Marazziti D. et al. (2009) [88]	18 OCD patients	-> CD8+ lymphocytes cells decreased after treatment	*p* = 0.004
		-> CD4+ lymphocytes increased after treatment	*p* = 0.005
Denys D. et al. (2006) [87]	42 OCD patients	No effect of paroxetine or venlafaxine on TNF-α, IL-4, IL-6, IL-10, IFN-γ, NK cell activity, monocytes, T-cells, and B-cells percentages	
Denys D. et al. (2004) [28]	Cf. cytokines AND (OCD OR “obsessive compulsive disorder”)		
Marazziti D. et al. (2003) [89]	10 OCD patients 10 controls	-> OCD patients (3)H-paroxetine-binding density < controls (3)H-paroxetine-binding density	*p* = 0.0001
Eisen JL. et al. (2001) [73]	Cf. antibody AND (OCD OR “obsessive compulsive disorder”)		
Murphy TK. et al. (2001) [74]	Cf. antibody AND (OCD OR “obsessive compulsive disorder”)		
Marazziti D. et al. (2001) [90]	10 OCD patients 15 controls	Presence of 5-HT2C and 5-HT2A mRNAs in patients and controls	
Rocca P. et al. (2000) [91]	15 OCD patients 10 controls	-> decrease of peripheral benzodiazepine receptor mRNA	*p* < 0.05
Marazziti D. et al. (1999) [76]	Cf. antibody AND (OCD OR “obsessive compulsive disorder”)		
Ravindran AV. et al. (1999) [92]	26 OCD patients 16 controls	-> OCD patients circulating NK cells < control circulating NK cell	*p* < 0.05
		No difference concerning B or T cells	
		No difference in circulating NK cells after treatment.	
Chapman F. et al. (1998) [77]	Cf. antibody AND (OCD OR “obsessive compulsive disorder”)		
Murphy TK. et al. (1997) [80]	Cf. antibody AND (OCD OR “obsessive compulsive disorder”)		
Swedo SE. et al. (1997) [81]	Cf. antibody AND (OCD OR “obsessive compulsive disorder”)		
Barber Y et al. (1996) [86]	7 OCD patients 9 controls	No difference in lymphocytes between OCD patients and OCD	
		No difference in lymphocytes after treatment.	
Rocca P. et al. (1991) [93]	18 OCD patients 50 controls	-> Number of binding sites for peripheral benzodiazepine receptor lower in OCD patients	*p* < 0.05
monocytes AND (“obsessive compulsive disorder” OR OCD)			
Rodriguez N et al. (2017) [35]	Cf. cytokines AND (OCD OR “obsessive compulsive disorder”)		
Denys D. et al. (2006) [87]	Cf. lymphocyte AND (“obsessive compulsive disorder” OR OCD)		
Denys D. et al. (2004) [28]	Cf. lymphocyte AND (“obsessive compulsive disorder” OR OCD)		
Weizman R. et al. (1996) [20]	Cf. cytokines AND (OCD OR “obsessive compulsive disorder”)		
NK cells” AND (“obsessive compulsive disorder” OR OCD)			
Denys D. et al. (2004) [28]	Cf. lymphocyte AND (“obsessive compulsive disorder” OR OCD)		
Ravindran V. et al. (1999) [92]	Cf. lymphocyte AND (“obsessive compulsive disorder” OR OCD)		

CD = cluster of differentiation; OCD = obsessive-compulsive disorder; HT2A = serotonin 2A; HT2C = serotonin 2C; IFN = interferon; mRNA = messenger ribonucleic acid; NK = natural killer; TNF = tumor necrosis factor.

**Table 4 brainsci-08-00149-t004:** Infectious agents and OCD.

Infection AND (OCD OR “Obsessive Compulsive Disorder”)			
Authors, Date	Subjects	Mains Results	Significance
Ursoiu F. et al. (2018) [94]	101 HIV patients	No association between HIV and OCD	
Akaltun I. et al. (2018) [52]	Cf. antibody AND (OCD OR “obsessive compulsive disorder”)		
Flegr J et al. (2017) [54]	Cf. antibody AND (OCD OR “obsessive compulsive disorder”)		
Sutterland AL. et al. (2015) [14] Meta-analysis	Cf. antibody AND (OCD OR “obsessive compulsive disorder”)		
Nicolini H. et al. (2015) [55]	Cf. antibody AND (OCD OR “obsessive compulsive disorder”)		
Miman O. et al. (2010) [63]	Cf. antibody AND (OCD OR “obsessive compulsive disorder”)		
Dale RC. et al. (2004) [95]	40 patients with post-streptococcal dyskinesias	-> 27.5% of these patients suffered from OCD	
Giulino L. et al. (2002) [96]	83 OCD patients	-> OCD patients with upper respiratory infection had more sudden onset than patients without upper respiratory infection	*p* = 0.02
		No difference concerning tic or ADHD comorbidity between OCD patients with or without upper respiratory infection.	
Lougee L. et al. (2000) [97]	54 PANDAS/OCD or tic patients 139 first relatives	-> 26% of OCD patients had a relative suffering from OCD	
lyme AND (“obsessive compulsive disorder” OR OCD)			
Johnco C. et al. (2018) [98]	147 patients with Lyme disease	-> 84% of patients reported obsessive compulsive symptoms	
		-> 90.9% of patients reported gradual onset of symptoms	
		-> 47% of patients were treated with psychotropic treatment and 76.9% of them presented at least partial improvement	
		-> 50.9% of patients treated with antibiotics reported at least partial improvement in symptoms	
Streptococcus AND (OCD OR “obsessive compulsive disorder”)			
Stagi S. et al. (2018) [99]	179 PANDAS/OCD or tic patients	-> reduced vitamin D in PANDAS patients	*p* < 0.0001
Mataix-Cols D. et al. (2017) [53]	Cf. antibody AND (OCD OR “obsessive compulsive disorder”)		
Calaprice D. et al. (2017) [100]	698 PANS patients	-> age of onset between 7 and 8 years	
		-> 88% of sudden onset	
		-> 87% of patients presented recurrences	
		-> 94% of patients presented a history of OCD	
		-> 71% with motor tics and 57% with vocal tics	
Wang HC. et al. (2016) [101]	2596 patients infected with GAS 25960 controls	-> increased risk of tic disorder in GAS infected patients	No full access
Nicolini H. et al. (2015) [55]	Cf. antibody AND (OCD OR “obsessive compulsive disorder”)		
Frankovich J. et al. (2015) [57]	Cf. antibody AND (OCD OR “obsessive compulsive disorder”)		
Ebrahimi Taj F. et al. (2015) [59]	Cf. antibody AND (OCD OR “obsessive compulsive disorder”)		
Murphy TK. et al. (2012) [61]	Cf. antibody AND (OCD OR “obsessive compulsive disorder”)		
Leckman JF. et al. (2011) [62]	Cf. antibody AND (OCD OR “obsessive compulsive disorder”)		
Murphy TK. et al. (2010) [102]	107 OCD or tic patients	-> 17.8% of patients had mother suffering from autoimmune disease	
Gause C. et al. (2009) [64]	Cf. antibody AND (OCD OR “obsessive compulsive disorder”)		
Kurlan R. et al. (2008) [103]	40 PANDAS/OCD or tic patients 40 non-PANDAS/OCD or tics	No difference in the number of exacerbations (but a strong tendency for increased exacerbation risk, *p* = 0.07).	*p* = 0.002
		-> more frequent GAS infection associated with exacerbation	
Dale RC. et al. (2004) [95]	Cf. infection AND (OCD OR “obsessive compulsive disorder”)		
Luo F. et al. (2004) [70]	Cf. antibody AND (OCD OR “obsessive compulsive disorder”)		
Pavone P. et al. (2004) [68]	Cf. antibody AND (OCD OR “obsessive compulsive disorder”)		
Murphy TK. et al. (2004) [69]	Cf. antibody AND (OCD OR “obsessive compulsive disorder”)		
Giulino L. et al. (2002) [96]	Cf. infection AND (OCD OR “obsessive compulsive disorder”)		
Murphy TK. et al. (2001) [74]	Cf. antibody AND (OCD OR “obsessive compulsive disorder”)		
Lougee L. et al. (2000) [97]	Cf. infection AND (OCD OR “obsessive compulsive disorder”)		
Giedd JN. et al. (2000) [104]	34 PANDAS/OCD or tics 82 controls	-> PANDAS patients mean caudate volume > controls mean caudate volume	*p* = 0.004
		-> PANDAS patients mean putamen volume > controls mean putamen volume	*p* = 0.02
		-> PANDAS patients mean globus pallidus volume > controls mean globus pallidus volume	*p* = 0.02
		No difference for thalamus and total brain volume	
Swedo SE. et al. (1998) [105]	50 PANDAS patients	-> Mean age at onset: 7.4 years	
		-> tics and OCD: 64%; tics only: 16% and OCD only: 20%	
		-> ADHD comorbidity: 40%, ODD comorbidity: 40%, MDD comorbidity: 36%	
Murphy TK. et al. (1997) [80]	Cf. antibody AND (OCD OR “obsessive compulsive disorder”)		
toxoplasma (OCD OR “obsessive compulsive disorder”)			
Akaltun I. et al. (2018) [52]	Cf. antibody AND (OCD OR “obsessive compulsive disorder”)		
Flegr J et al. (2017) [54]	Cf. antibody AND (OCD OR “obsessive compulsive disorder”)		
Sutterland AL. et al. (2015) [14]	Cf. antibody AND (OCD OR “obsessive compulsive disorder”)		
Miman O. et al. (2010) [63]	Cf. antibody AND (OCD OR “obsessive compulsive disorder”)		

ADHD = attention deficit/hyperactivity disorder; GAS = Group A streptococcus; HIV = human immunodeficiency virus; MDD = major depressive disorder; ODD = oppositional defiant disorder; PANDAS = pediatric autoimmune neuropsychological disorders associated with streptococcal infection; PANS = pediatric acute-onset neuropsychiatric syndrome.

**Table 5 brainsci-08-00149-t005:** Specific immunological treatment in OCD.

(PANDAS OR PANS) AND Treatment AND (OCD OR “Obsessive Compulsive Disorder”)			
Authors, Date	Subjects	Main Results	Significance
Leon J. et al. (2018) [118]	33 PANDAS patients	Follow-up lasted between 2.2 and 4.8 years	
		Initially, all patients treated with antibiotics	
		During the follow-up period, 45% of patients took psychotropic treatments	
		At the time of follow-up, 18 patients presented no symptoms, 11 only subclinical symptoms, 3 moderate symptoms, and 1 severe symptom.	
Calaprice D. et al. (2018) [120]	698 PANS patients	675 patients treated with antibiotics, 437 with anti-inflammatories, 378 with psychotropic treatments	
		52% of “very effective” treatments with antibiotics	
		NSAIDs were at least “somewhat effective” for 80% of patients	
		Steroids were at least “somewhat effective” for 72% of patients	
		IVIG were at least “somewhat effective” for 74% of patients	
Brown K. et al. (2017) [121]	98 PANS patients	-> duration of symptomatic periods treated with steroids < duration of symptomatic periods of non-treated patients	*p* < 0.001
		-> shorter symptomatic periods when initially treated with steroids	*p* < 0.01
Brown KD. et al. (2017) [126]	95 PANS patients	-> Symptomatic periods treated with NSAID lasted shorter than non-treated symptomatic periods	*p* < 0.0001
		-> the more the duration without treatment is short, the more symptomatic period were short	*p* = 0.02
Spartz EJ. et al. (2017) [122]	159 PANS patients	No clinical data allow to distinguish responders and non-responders to NSAIDs	
		31% of patients with NSAID increases reported improvement in symptoms	
		35% of patients with NSAID removal reported symptom increases after removal	
Murphy TK. et al. (2017) [123]	31 PANS patients (17 with azithromycin, 14 with placebo)	-> azithromycin group improvement > non-azithromycin group (CGI)	*p* = 0.003
		No difference on CY-BOCS	
Calaprice D. et al. (2017) [100]	Cf. Streptococcus AND (OCD OR “obsessive compulsive disorder”)		
Williams KA. et al. (2016) [124]	35 PANDAS patients (IVIG group = 17, placebo group = 18)	-> At week 6 (double blind phase): no difference between groups (CY-BOCS)	*p* < 0.0001
		-> Improvement after open label IVIG (CY-BOCS)	
Nadeau JM. et al. (2015) [127]	11 PANS patients partially responder to antibiotics	-> Improvement after CBT (CY-BOCS)	*p* = 0.01
Nicolini H. et al. (2015) [55]	Cf. antibody AND (OCD OR “obsessive compulsive disorder”)		
Frankovich J. et al. (2015) [57]	Cf. antibody AND (OCD OR “obsessive compulsive disorder”)		
Latimer ME. et al. (2015) [128]	35 PANDAS patients	-> 6 months after therapeutic plasma apheresis: improvement of 65% (local questionnaire)	
Demesh D. et al. (2015) [129]	10 PANDAS patients	-> Improvement in symptom intensity after antibiotic treatment (local questionnaire)	*p* = 0.03
		-> Improvement in symptom intensity after tonsillectomy (local questionnaire)	*p* = 0.02
Ebrahimi Taj F. et al. (2015) [59]	Cf. antibody AND (OCD OR “obsessive compulsive disorder”)		
Pavone P. et al. (2014) [130]	120 PANDAS patients (56 patients with tonsillectomy or adrenotonsillectomy, 64 without)	No difference concerning symptomatology, streptococcal antibodies or anti-neural antibodies (evaluation every two months for 2 years)	
Murphy TK. et al. (2012) [61]	Cf. antibody AND (OCD OR “obsessive compulsive disorder”)		
Bernstein GA. et al. (2010) [131]	21 PANDAS patients 18 non-PANDAS OCD patients	No difference concerning age at onset of OCD	
		No difference concerning CY-BOCS score	
		-> PANDAS patients YGTSS score > non-PANDAS patients YGTSS score	*p* = 0.013
		No difference concerning ASO or anti-DNase B titers	
		-> In non-PANDAS OCD patients, separation anxiety disorder and social phobia are more frequent	*p* = 0.02 and 0.047 respectively
Storch EA. et al. (2006) [132]	7 PANDAS patients	-> CY-BOCS improvement after 3 weeks of CBT	*p* = 0.018
Snider LA. et al. (2005) [133]	23 PANDAS patients	-> Decrease in number of symptom exacerbations with antibiotic treatment	*p* < 0.01
Garvey MA. et al. (1999) [134]	37 PANDAS patients (double blind and cross over design)	No difference in symptoms following antibiotic treatment	
Swedo SE. et al. (1998) [105]	Cf. Streptococcus AND (OCD OR “obsessive compulsive disorder”)		
NSAID and (OCD OR “obsessive–compulsive disorder”)			
Brown KD. et al. (2017) [126]	Cf. (PANDAS OR PANS) AND treatment AND (OCD OR “obsessive compulsive disorder”)		
Spartz EJ. et al. (2017) [122]	Cf. (PANDAS OR PANS) AND treatment AND (OCD OR “obsessive compulsive disorder”)		
Shalbafan M. et al. (2015) [135]	25 OCD patients with celecoxib (+SRI) 25 OCD patients with placebo (+SRI)	-> lower CY-BOCS score at week 10 in celecoxib group than in placebo group	*p* = 0.047
Sayyah M. et al. (2011) [136]	27 OCD patients with celecoxib (+fluoxetine) 25 OCD patients with placebo (+fluoxetine)	-> lower CY-BOCS score at week 8 in celecoxib group than in placebo group	*p* = 0.037
		-> significant effect of time-by-treatment interaction in ANOVA	*p* = 0.018
“anti-inflammatory” and (OCD OR “obsessive–compulsive disorder”)			
Calaprice D. et al. (2018) [120]	Cf. (PANDAS OR PANS) AND treatment AND (OCD OR “obsessive compulsive disorder”)		
Brown K. et al. (2017) [121]	Cf. (PANDAS OR PANS) AND treatment AND (OCD OR “obsessive compulsive disorder”)		
Brown KD. et al. (2017) [126]	Cf. (PANDAS OR PANS) AND treatment AND (OCD OR “obsessive compulsive disorder”)		
Shalbafan M. et al. (2015) [135]	Cf. NSAID and (OCD OR “obsessive–compulsive disorder”)		
Sayyah M. et al. (2011) [136]	Cf. NSAID and (OCD OR “obsessive–compulsive disorder”)		
minocycline and (OCD OR “obsessive–compulsive disorder”)			
Esalatmanesh et al. (2016) [137]	47 OCD patients with minocycline (+fluvoxamine) 47 OCD patients with placebo (+fluvoxamine)	-> lower Y-BOCS score at week 10 in minocylcine group than in placebo group	*p* = 0.008
Rodriguez CI. et al. (2010) [138]	9 OCD patients with minocycline (+SRI)	No effect of minocycline at week 12	
N-acetylcysteine and (OCD OR “obsessive–compulsive disorder”)			
Ghanizadeh A. et al. (2017) [139]	18 OCD patients with NAC (+citalopram) 11 OCD patients with placebo (+citalopram)	-> lower Y-BOCS score at week 12 in NAC group than in placebo group	*p* < 0.02
Costa DLC. et al. (2017) [140]	40 OCD patients randomized in 2 groups: NAC and placebo (no access to the details)	-> No difference between the two groups concerning Y-BOCS scores.	
Paydary K. et al. (2016) [141]	23 OCD patients with NAC (+fluvoxamine) 23 OCD patients with placebo (+fluvoxamine)	-> No difference between the two groups concerning Y-BOCS at week 10.	
Sarris J. et al. (2015) [142]	22 OCD patients with NAC (+TAU) 22 OCD patients with placebo (+TAU)	-> No difference between the two groups concerning Y-BOCS at week 16.	
Afshar F. et al. (2012) [143]	24 OCD patients with NAC (+SRI) 24 OCD patients with placebo (+SRI)	-> lower Y-BOCS score at week 12 in NAC group than in placebo group	*p* = 0.03

ASO = anti-streptolysin O; CBT = cognitive behavioral therapy; CGI = clinical global impression; CY-BOCS = Children’s Yale–Brown Obsessive Compulsive Scale; IVIG = intravenous immunoglobulin; NAC = N-acetylcysteine; NSAID = non-steroidal anti-inflammatory drug; PANDAS = pediatric autoimmune neuropsychological disorders associated with streptococcal infection; PANS = pediatric acute-onset neuropsychiatric Syndrome; SRI = serotonin reuptake inhibitor; TAU = treatment as usual; Y-BOCS = Yale-Brown Obsessive Compulsive Scale; Y-GTSS = Yale Global Tic Severity Scale.

**Table 6 brainsci-08-00149-t006:** Summary.

Divergent results concerning cytokines (especially IL-6, TNF-α) were found between studies. These discrepancies, therefore, raise the question of different patient populations, with some patients possibly presenting with immunological deficiencies, thus explaining the discrepancies.
Antibody studies show that autoimmune factors could be specific etiologies in OCD.
*Streptococcus pyogenes* is already recognized as possibly leading to OCD through PANS (pediatric acute-onset neuropsychiatric syndrome), as is *Toxoplasma gondii*. The mechanisms leading to OCD for *S. pyognes* and *T. gondii* are still unknown, but autoimmunity seems to be involved.
According to these different possible immune etiological factors (autoimmunity, infection), some specific treatments were already tested opening the way to individualized specific treatments. An effort to clearly distinguish between the different etiological (including immunological) factors is still necessary in order to develop more effective OCD treatments

IL = interleukin; OCD = obsessive-compuslve disorder; PANS = pediatric acute-onset neuropsychiatric syndrome; TNF = tumor necrosis factor.

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
