# Peer review of "Individualized Immunological Data for Precise Classification of OCD Patients"

_brainsci, 2018, doi:10.3390/brainsci8080149_

Round 1

Reviewer 1 Report

Although I understand that this is a narrative and not a systematic review, we still need to see a methods section, that describes the search strategy. It needs to cover search terms, data bases, date, strategy to screen the relevant papers, etc.

There is a need to bring some of the literature from other disorders. Several other disorders such as depression and anxiety are described using a theory of infectious disease. This section could suggest that it is not just OCD but several other conditions that have the same explanation.

Implications are missing. Implications should go beyond future research. Can clinicians do anything if the disorder has an infectious etiology? What exact areas need research. What are the major weaknesses and what is the future directions. Please guide the future research.

Are there studies that social class, poverty, or stress may mediate or confound this theory? Stress reduces immune response. Poor people also have weaker immune function. They may also have differential exposure to germs. Please describe how other factors may cause OCD through infections.

What about other infections? These two studied infections are not the only studied infectious aetiologies. Cover some additional ones.

A review paper needs a table that summarizes the results, weaknesses, sample size, and sample of previous studies. Add such a table.

Author Response

As tracked change function made the text difficult to read, we posted our changes with bold text. You will find the corrected manuscript in attachment.

Reviewer 1:

Although I understand that this is a narrative and not a systematic review, we still need to see a methods section, that describes the search strategy. It needs to cover search terms, data bases, date, strategy to screen the relevant papers, etc.

Response: We thank the reviewer for this useful comment. We have created a methods section where we describe our search strategy: use of Pubmed database; key words. We detailed the results obtained with these keywords and the number of finally selected papers, detailing our inclusion and exclusion criteria for the papers found. Finally, we present the final results in different tables.

You will find modification in method section (page 3 - 22, line 67 - 125 in the corrected version), with tables (page 3 – 21, line 108 – 125 in the corrected version).

Furthermore, the systematization of my work led me to modify the text with more details:

-       Page 22, line 134 – 136 in the corrected version, in bold: “Further studies [29-35,37,38] have been carried out enabling a meta-analysis [28] that found decreased IL-1β level, decreased TNFα levels in non-depressed OCD patients (but not in OCD patients with possible comorbid depression) and increased IL-6 level in adult medication-free OCD patients (but not in OCD children with possible medication use)compared to controls”

-       Page 23, line 170 – 171 in the corrected version: “Most studies concerning antibodies in OCD concerns pediatric autoimmune neuropsychological disorders associated with streptococcal (PANDAS) infections”

-       Page 23, line 174-176 in the corrected version: “These studies strongly support the existence of an auto-immune etiological factor in OCD. However, discrepancies still exist: ABGAs have been found in OCD patients (and not in controls) but not in all OCD patients [55].”

-       Page 24, line 235 – 236 in the corrected version: “As in Sydenham’s chorea, anti-lysoganglioside antibodies seem involved [47,56].”

-       Page 24, line 247 in the corrected version: “This result of a good prognostic is confirmed by Murphy et al. [50] but not by Frankovich et al. [46].”

-       Page 26, line 327 – 329 in the corrected version, in bold: “Minocycline has been studied as a potential new pharmacological treatment for OCD, and the results are mixed: one study found that minocycline could be a good adjunctive treatment to classical OCD treatment with SSRIs [113] but another one did not find this result [114] »

-       Page 27, line 388 – 390 in the corrected version: “The identification of more specific biological clusters in OCD is essential in order to advance our knowledge and treatment of OCD.”

There is a need to bring some of the literature from other disorders. Several other disorders such as depression and anxiety are described using a theory of infectious disease. This section could suggest that it is not just OCD but several other conditions that have the same explanation.

Response: We agree with this comment, but as we decided to conduct a systematic review on OCD, we thought that it would be off topic to devote a specific paragraph to other disorder. However, we have added the following in the introduction section: “This question of a possible infectious etiology has also been suspected in other psychiatric disorders [14].”,

(page 2, line 56 - 57 in the corrected version)

Implications are missing. Implications should go beyond future research. Can clinicians do anything if the disorder has an infectious etiology? What exact areas need research. What are the major weaknesses and what is the future directions. Please guide the future research.

Response: We have described, in the therapeutic section, what a clinician could do in the case of infectious etiology of the disorder. For example in the case of PANDAS, page 26, line 312 - 315 in the corrected version, “However, if streptococcal infection is still present during acute episodes of PANDAS, antibiotics are considered as the best treatment [164,165]. Finally, corticosteroid and nonsteroidal anti-inflammatory (NSAIDs) drugs do appear to be effective [97-100]”. Unfortunately, specific treatments in the case of other specific infections are not yet known.

In the conclusion section, we suggest some research priorities in this field. Specifically, we explain how it might be important to determine specific biological OCD clusters to better treat our patients and to better understand OCD biology:  page 27, line 386 – 390, in the corrected version, “Future research should focus on these etiological factors (genetic, immunological, etc.) in order to elucidate the biological bases of OCD, and develop prevention tools and better treatments [189], to pave the way to precision individualized therapies [190], for the benefit of patients. The identification of more specific biological clusters in OCD is essential in order to advance our knowledge and treatment of OCD.”

Are there studies that social class, poverty, or stress may mediate or confound this theory? Stress reduces immune response. Poor people also have weaker immune function. They may also have differential exposure to germs. Please describe how other factors may cause OCD through infections.

Response: We agree with the reviewer that poverty, social class or stress could possibly be confounding factors in the causality of immune factors in OCD. Yet, in the literature, if auto-immune or infectious factors as well as immune gene disfunction are evoked as possible etiologies factor of subtypes of OCD, no study to our knowledge has assessed the role of social factors.

What about other infections? These two studied infections are not the only studied infectious aetiologies. Cover some additional ones.

Responses: We have found possible HIV or Lyme etiologies in OCD, and we have mentioned these studies (Table 4, ref 84 and 88, page 13 and 14 in the corrected version).

A review paper needs a table that summarizes the results, weaknesses, sample size, and sample of previous studies. Add such a table.

Response: We thank the reviewer for this comment. We added tables summarizing these points in the method section (Table 1 to 6, page 3 – 22, line 108 - 124 in the corrected version, and page 26, line 335 in the corrected version).

Reviewer 2 Report

The authors present a qualitative, narrative review of immunological factors in OCD. A few points to consider in revisions:

-Some grammar and spelling errors were identified throughout. Please re-check for errors if possible.

-The abstract should give some sort of narrative or brief conclusions regarding the data your are giving an overview of within this manuscript.

- Consider providing a brief overview of the cytokines that you discuss given that this paper could be read by clinicians that may not be as familiar with these pathways.

-The paper could benefit from several tables throughout summarizing your review. Summarize studies that support the possible role for s. pyogenies and tox gondii in OCD.

-Additionally, the pharmacotherapy/alternative treatment section could be expanded on and summarized in a table format as well. More detail is needed on medications that have been tried and their potential efficacy (as well as side effects that limit their utility). You also mention certain therapies and don't provide much information on them such as tocilizumab in line 80 is not referenced in the therapy section.

Author Response

As tracked change function made the text difficult to read, we posted our changes with bold text.

Reviewer 2:

The authors present a qualitative, narrative review of immunological factors in OCD. A few points to consider in revisions:

-The abstract should give some sort of narrative or brief conclusions regarding the data you are giving an overview of within this manuscript.

Response: In line with this comment, we rewrote the abstract (page 1, line 17 – 27 in the corrected version): “We found discrepancies concerning cytokines, raising the hypothesis of specific immunological etiological factors. Antibody studies confirm this hypothesis showing a potential auto-immune etiological factor”.

- Consider providing a brief overview of the cytokines that you discuss given that this paper could be read by clinicians that may not be as familiar with these pathways.

Response: We thank the reviewer for this comment. We have inserted a paragraph on this topic page 22, line 150 - 162 in the corrected version:

“TNFα, IL-1β and IL-6 are inflammatory cytokines (for a very complete review, see [121]). TNFα is produced by a wide range of cells including T or B cells and monocytes (including microglia) and targets all nucleated cells. TNFα has a complex role, being both pro-inflammatory and immunosuppressive. In the brain, TNFα could be involved in synapses scaling with high level of TNFα favoring LTP and low level of TNFα favoring LTD [123,124]. Progranulin mutation has been found to be associated with hyperexcitability of nucleus accumbens spiny neurons in mice, in line with hyperactivity of cortico-striatal loops in OCD [1], and elevated TNFα level and hyperactivation of microglia [125]. With the progranulin gene restored, OCD-like behavior disappeared in mice [125]. Frontoparietal dementia patients showing mutations of progranulin present OCD [125]

IL-1β is also produced by microglia and targets T cell or endothelial and epithelial cells [121].  IL-6 is produced by both astrocytes and microglia, and IL-6 exposure could increase synaptic activity (for an excellent review on IL-6 central nervous system effects see [126]).”

-The paper could benefit from several tables throughout summarizing your review. Summarize studies that support the possible role for s. pyogenies and tox gondii in OCD.

Response: In line with this comment and also the comment of reviewer 1, we have added tables in which, in particular, studies supporting the putative role of Streptococcus and Toxoplasma are described (table 4, pages 13 and 16, line 115 – 118 in the corrected version)

-Additionally, the pharmacotherapy/alternative treatment section could be expanded on and summarized in a table format as well. More detail is needed on medications that have been tried and their potential efficacy (as well as side effects that limit their utility). You also mention certain therapies and don't provide much information on them such as tocilizumab in line 80 is not referenced in the therapy section.

Response: We have described studies of the alternative treatments tried. These are summarized in the additional tables in the methods section (table 5, pages 16 to 22 line 118 – 124 in the corrected version).

Concerning tocilizumab, it is not a treatment for OCD. We therefore did not mention it in the tables. We have simply hypothesized about IL-6 involvement in OCD and the implications this could have at a therapeutic level. For this reason, tocilizumab is not referenced further below, in the therapy section.

Round 2

Reviewer 1 Report

The revision is satisfactory. a table is added. more studies are included. similar psych disorders are mentioned. more methods info is provided, overall he paper is much better. 

Reviewer 2 Report

The authors have satisfactorily revised their manuscript according to reviewer comments